# Geriatric ocular trauma and mortality: A retrospective cohort study

**Vincent Q. Pham**[1], **Hannah M. Miller**[2], **Elise O. Fernandez**[2], **Daniel de Marchi**[3], **Elizabeth Budi**[4], **Hongtu Zhu**[3], **David Fleischman**[2]*

1 Northeast Ohio Medical University, Rootstown, Ohio, United States of America, 2 Kittner Eye Center, University of North Carolina Hospitals, Chapel Hill, North Carolina, United States of America, 3 Department of Biostatistics, University of North Carolina at Chapel Hill, Chapel Hill, North Carolina, United States of America, 4 University of California Irvine, Irvine, California, United States of America

* david_fleischman@med.unc.edu

## Abstract

### Purpose

The objective of this investigation is to evaluate the 5-year mortality of geriatric patients who have sustained eye injuries.

### Design

This retrospective cohort study included patients aged 65 years or older who had histories of either ocular trauma or age-related nuclear cataracts.
Subjects and controls: Patients with ocular trauma constituted the study group, while those with a history of cataracts served as controls.

### Methods

Data from the I2B2 Carolina Data Warehouse were analyzed. Patient demographics were collected, and the outcomes of interest were the overall mortality rate and annual mortality rates over a 5-year period. Chi-squared tests were utilized for the comparison of mortality data.

### Main outcomes and measures

The primary outcomes were overall mortality rates and annual mortality rates expressed as percentages.

### Results

The study group consisted of 602 patients who had suffered ocular trauma. The control group included 1066 patients of similar age who had been diagnosed with age-related nuclear cataracts at some point in their lives. Among the study group, 74 patients died within 5 years, while 69 patients in the control group died within the

**Data availability statement:** Data Availability Statement for PLOS ONE Most of the relevant data are within the manuscript and its supporting files. Our submission contains most of the raw data needed to replicate the results of the study. While we have tried to share the "minimal data set" which is the data required to replicate all study findings in the articles, certain aspects of our code and data cannot be shared due to legal or ethical constraints which would breach parameters that were defined and set in place prior to the start of the project. Our code and data contain sensitive legal and ethical data such as potentially identifying or sensitive patient information that cannot be fully shared publicly. However, I will provide various UNC e-mails that could be contacted to view the code and additional data points. Please feel free to contact the following Sandy Barnhart: Social Research Specialist, e-mail: sandy_barnhart@med.unc.edu I2B2 and NCTracs Support Specialist: Joe Mosnier, mosnier@unc.edu and nctracs@unc.edu

**Funding:** The project described was supported by the National Center for Advancing Translational Sciences (NCATS), National Institutes of Health (UL1TR002489) for author VP. The content is solely the responsibility of the authors and does not necessarily represent the official views of the NIH. The research reported in this publication was supported by the National Institute on Aging, of the National Institutes of Health (NIA 2-T35-AG038047, T35AG038047) through the UNC-CH Summer Research Training in Aging for Medical Students for author VP. The content is solely the responsibility of the authors and does not necessarily represent the official views of the NIA or NIH.

**Competing interests:** The authors have declared that no competing interests exist.

same timeframe, resulting in a study group mortality rate of 11.30% and a control group mortality rate of 6.47%. For patients with ocular trauma, the annual mortality rates were 4.15%, 2.60%, 1.96%, 2.54%, and 0.56%, respectively. For the control group, the annual mortality rates were 1.03%, 1.70%, 1.64%, 0.88%, and 1.38% respectively.

## Conclusion

The study suggests that geriatric patients who have experienced ocular trauma are at a higher risk of mortality compared to age-matched controls without such injuries. These findings highlight the necessity of identifying the causes of geriatric periorbital trauma and underscore the importance of close patient follow-up to improve outcomes.

## Introduction

Ocular trauma is a leading cause of vision loss and visual disability, contributing to 65% of unilateral blindness cases [1]. It accounts for up to 52% of ophthalmic accidents and emergency cases and is a significant cause of ocular morbidity [2]. Falls, a predominant cause of injury in the geriatric population, may lead to death, morbidity, or damaged periocular tissue [2]. Current research on geriatric ocular trauma is limited, as is a well-defined system to classify orbital and adnexal trauma. For the purposes of this paper, 'ocular trauma' will encompass globe, orbital, and adnexal injuries. This study focuses on mortality rates associated with traumatic eye injuries as compared to non-traumatic cataract patients.

Orbital fractures, a common form of ocular trauma, can occur in any of the bones forming the orbit and often result from blunt force trauma. These fractures may lead to entrapment of extraocular muscles, enophthalmos, and decreased ocular motility [3]. Among the most common injuries from facial trauma, orbital fractures can lead to debilitating ocular dysfunction [4,5]. Such fractures are prevalent in the geriatric population, especially among those with complex medical histories [4].

Periorbital trauma, a more recent classification of ocular injury, can affect both the superficial and deep parts of the orbit as well as periocular, frontal, temporal, and malar regions [6]. Injuries may include damage to the levator palpebrae superioris, orbital fracture, skin lacerations, tearing of the medial canthus, or conjunctival damage [6]. The onset of functional deficits such as ptosis and enophthalmos may occur thereafter [6]. Falls may also cause full-thickness rupture or penetrating injury to the eye itself, resulting in permanent vision loss.

Falls are linked to increased mortality rates among the elderly, who often sustain intracranial injuries as a result [7]. Geriatric patients are susceptible to head injuries due to diminished reflexes, movement speed, and muscle strength associated with aging [7]. Furthermore, advancements in technology and medical treatments have enabled longer lifespans, but often with multiple chronic conditions. These conditions, along with physiologic impairments, can hinder older patients' response to

emergencies such as falls. While numerous studies have explored the association between falls, mortality, open globe injuries, and visual outcomes following eye injuries [7,8], there is a notable gap in research focusing on ocular trauma as a risk factor for geriatric mortality.

Falls are a leading cause of morbidity in the elderly. Over 33% of individuals aged 65 years or older fall annually, a figure that doubles when cognitive impairment is present [9]. To minimize falling risks, patients need a normal gait and efficient neurosensory processing [9]. Aging commonly leads to balance impairment, decreased motor strength, and visual impairments, increasing the risk of falls [9]. Additional risk factors for falls in geriatric patients include environmental hazards, frailty, vestibular disorders, cognitive impairment, and side effects from medications [10]. For a fall to result in ocular trauma, there must be a failure in the protective reflexes that guard the face, as it is instinctive to shield the eyes and head during a fall. We hypothesize that ocular trauma from falls may signify early neurologic and systemic decline in elderly patients. Neurodegeneration can impact executive functions and gait, thereby increasing fall propensity in the elderly, particularly those with dementia [11]. This study investigates patients with periorbital injury, orbital fractures, and ocular trauma who visit healthcare facilities within 5 years post-injury, focusing on their mortality rates in comparison with non-traumatic cataract patients.

## Materials & methods

This retrospective cohort study utilized data from the I2B2 University of North Carolina (UNC) database, offering an overview of patients within the UNC Health Care system and enabling researchers to query aggregate data. The dataset included patients aged 65 and older evaluated at UNC hospitals between April 2011 and June 2016, with a minimum of five years of follow-up. To ensure comparability despite sample size differences, we performed a secondary covariate-matched analysis, demonstrating consistent mortality trends. The study group consisted of patients who sustained ocular trauma. The control group comprised patients of the same age range diagnosed with nuclear sclerotic cataracts at any point, with a minimum follow-up of 5 years or until the time of death.

We applied a filter and search for patients 65 or older meeting the inclusion criteria of ocular trauma with a 5-year follow-up from the time of presentation or a documented time of death. The study group presented with injuries such as those to the eye and orbit, open wounds of the eyelid and periocular area, fractures of the orbital floor, facial bones, injury to the conjunctiva, contusion of the eyeball, ocular laceration, penetrating wounds of the orbit, or avulsion of the eye. The control group consisted of patients with age-related nuclear cataracts. We excluded those with multisystem trauma, victims of transport accidents, homicide, assault, vehicle accidents, intentional injuries, or those who died within one month of presentation to minimize bias in mortality rate statistics. Exclusions aimed to remove deaths due to external trauma rather than systemic health decline. Sensitivity analysis confirmed that excluding these cases did not significantly impact the study's mortality trends. Further details on inclusion and exclusion criteria are available in "Fig 1". The resulting study and control groups were collectively designated as the pre-matching group.

Python was used to code various data aspects, including mortality and diagnoses. We calculated the overall mortality rate and the mortality rates at 1, 2, 3, 4, and 5 years, detailed in "Table 1". Statistical analyses evaluated mortality by age, decade, and sex.

Data were further segmented into post-matching groups to examine mortality through a system-based perspective, particularly patient comorbidities. We conducted 1:1 covariate matching based on initial diagnoses at presentation to align study and control groups with comparable baseline characteristics, allowing for controlled comparison of mortality data across cardiovascular and neurological diseases. We matched patients in the study group and control group using a nearest-neighbor algorithm based on various systemic, neurologic, and cardiovascular comorbidities considering diseases such as heart disease, vascular disease, hypertension, neurodegenerative disease, dementia, etc. at baseline. We also considered patient age during this process, resulting in a robust post-match patient sample. Further details on covariate matching are available in "Fig 1". These disease categories for covariate matching are outlined in "Table 2".

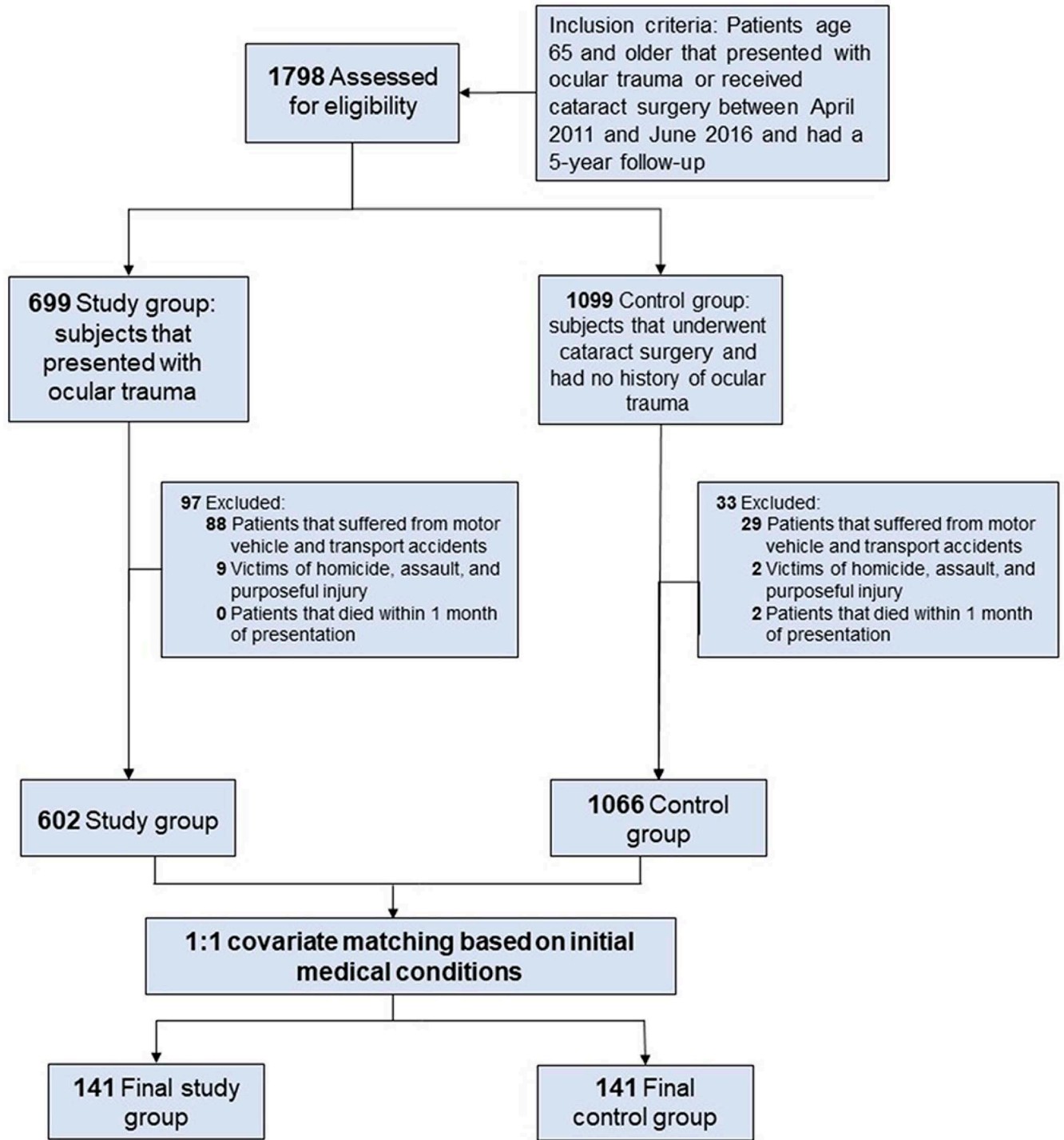

**Fig 1. Exclusion Criteria and Covariate Matching.** After exclusion, 602 patients had ocular trauma, whereas 1066 patients had a history of cataracts. Both groups were assessed within a 5-year follow-up period. Covariate matching was performed based on initial medical conditions including cardiovascular and neurological conditions resulting in 141 patients in the final study group and 141 patients in the final control group.

**Table 1. Mortality rates of control vs study in Pre-Matching Group.**

|  | Control Group | Study Group | P-value |
|---|---|---|---|
| # of Patients | 1066 | 602 |  |
| # of deaths | 69 | 68 |  |
| Overall Mortality | 6.47% | 11.3% | 0.0056 |
| 1 Yr Mortality | 1.03% | 4.15% | <0.0001 |
| 2 Yr Mortality | 1.71% | 2.6% | 0.21498 |
| 3 Yr Mortality | 1.64% | 1.96% | 0.63122 |
| 4 Yr Mortality | 0.88% | 2.54% | 0.00694 |
| 5 Yr Mortality | 1.38% | 0.56% | 0.12114 |

There is a noticeable increase in 1-year mortality in the study group which contains patients with ocular trauma.

**Table 2. List of cardiovascular and neurological morbidities evaluated in post-matching groups.**

| Cardiovascular diseases | Neurologic diseases | Other |
|---|---|---|
| Cardiac arrhythmias<br>Ischemic heart diseases<br>Heart failure<br>Atherosclerosis, aortic aneurysm, peripheral vascular disease, atheroembolism or disease of capillaries<br>Hypertensive heart diseases including essential primary hypertension, hypertensive heart disease, secondary hypertension, hypertensive crisis | Dementia<br>Degenerative diseases of nervous system<br>Polyneuropathies and other disorders of peripheral nervous system not from infectious disease<br>Encephalopathy<br>Alzheimer's disease<br>Parkinson's disease | Type 2 diabetes mellitus |

Cardiovascular comorbidities encompassed cardiac arrhythmias, type 2 diabetes, ischemic heart disease, heart failure, vascular disease, and hypertensive heart diseases. Neurological comorbidities included dementia, degenerative diseases of the nervous system, polyneuropathies, encephalopathy, Alzheimer's disease, and Parkinson's disease. Diagnoses were evaluated using North Carolina I2B2 guidelines. So, if the patient presented to the hospital before 2015, we would use ICD-9 codes to index the diagnosis. For patients that presented after 2015, we used ICD-10 codes to index the diagnosis. Mortality was computed across patient sex and age decade.

Demographics, surgical history, insurance status, ophthalmic history, injury timing, comorbidities, and mortality were extracted and coded according to the International Classification of Diseases (ICD) guidelines. Diagnoses and procedures were coded in ICD-9 format for presentations before 2015 and in ICD-10 format thereafter. We incorporated both ICD-9 and ICD-10 coding systems to ensure comprehensive data collection. Using the Carolina Data Warehouse for Health as an initial data source, we developed a Python script on Jupyter Notebook to extract relevant patient data and perform analyses. We accessed this data on September 17, 2021, and again on January 1, 2022. Institutional Review Board (IRB) and Ethics Committee approval were secured in line with UNC guidelines, adhering to the Declaration of Helsinki.

We utilized two-sided chi-squared tests to compare the mean mortality rates of male and female patients presenting with ocular trauma. The alpha level was set at 0.05 for significance testing, with a Bonferroni correction to maintain a family-wise error rate of 0.05, resulting in an adjusted alpha level of 0.025 for individual tests. The hypothesis testing was conducted using Scipy 1.9 within the Python 3.7 environment. In addition to chi-squared tests, we performed a Cox proportional hazards regression adjusting for age, sex, cardiovascular conditions, and neurological conditions. Results confirmed the significant association between ocular trauma and increased mortality risk.

 

## Results

To mitigate potential bias, patients who were victims of transport accidents, homicide, assault, vehicle accidents, intentional injury, or who died within 1 month of presentation were excluded. Consequently, 97 patients were removed from the study group and 33 from the control group. Detailed inclusion and exclusion criteria are depicted in "Fig 1".

### Patient demographics

The pre-matching group encompassed 1668 patients: 602 in the study group and 1066 in the control group. Within the control group, 624 (58.5%) were female and 442 (41.4%) were male, with a mean age of 75.6 and a median age of 75. Racial demographics revealed 745 (69.8%) white, 212 (19.8%) black or African American, 29 (2.7%) Asian, 53 (4.9%) of other races, and 23 (2.1%) unknown. In the study group, 391 (64.9%) were female and 211 (35.0%) were male, with a mean age of 80.7 years and a median age of 80. This group included 502 (83.3%) white, 64 (10.6%) black or African American, 10 (1.6%) Asian, 11 (1.8%) of other races, and 14 (2.3%) with unknown racial backgrounds. Additional demographic information is provided in "Table 3".

The post-matching group was formed through covariate matching and comprised 141 patients in both the study and control groups. See "Fig 1" for further details. In the study group, 93 (65.9%) were female and 48 (34.0%) were male, with a mean age of 81.8 and a median age of 81. Among these, 114 (80.8%) were white, 22 (15.6%) were black, 2 (1.4%)

Table 3. Patient demographics of subjects in pre-matching and post-matching groups.

| Patient Demographics | Pre-matching control group (n = 1066) (%) | Pre-matching study group (n = 602) (%) | Post-matching control group (n = 141) (%) | Post-matching study group (n = 141) (%) |
|---|---|---|---|---|
| **Age** | | | | |
| Mean age | 75.6 | 80.7 | 76.2 | 81.8 |
| Median Age | 75 | 80 | 76 | 81 |
| Minimum Age | 61 | 65 | 64 | 68 |
| Maximum Age | 100 | 103 | 100 | 102 |
| **Sex** | | | | |
| Female | 624 (58.5) | 391 (64.9) | 87 (61.7) | 93 (65.9) |
| Male | 442 (41.4) | 211 (35.0) | 54 (38.2) | 48 (34.0) |
| **Race** | | | | |
| White | 745 (69.8) | 502 (83.3) | 105 (74.4) | 114 (80.8) |
| Black or African American | 212 (19.8) | 64 (10.6) | 29 (20.5) | 22 (15.6) |
| Asian | 29 (2.7) | 10 (1.6) | 2 (1.4) | 2 (1.4) |
| Native Hawaiian or other Pacific Islander | 0 (0) | 0 (0) | 1 (0.7) | 0 (0) |
| Other | 53 (4.9) | 11 (1.8) | 3 (2.1) | 2 (1.4) |
| Unknown | 23 (2.1) | 14 (2.3) | 1 (0.7) | 1 (0.7) |
| **Ethnicity** | | | | |
| Non-Hispanic | 996 (93.4) | 576 (95.6) | 137 (97.1) | 138 (97.8) |
| Hispanic | 39 (3.6) | 6 (0.9) | 2 (1.4) | 2 (1.4) |
| Unknown | 26 (2.4) | 17 (2.8) | 2 (1.4) | 1 (0.7) |
| **Insurance Status** | | | | |
| Medicare/Medicaid/State Government | NA | NA | 128 (90.7) | 134 (95.0) |
| Private | NA | NA | 8 (5.6) | 2 (1.4) |
| Self-Pay | NA | NA | 5 (3.5) | 5 (3.5) |

NA: Not available.

were Asian, 2 (1.4%) of other racial backgrounds, and 2 (0.7%) were of unknown race. The control group included 87 (61.7%) female and 54 (38.2%) male patients, with a mean age of 76.2 and a median age of 76, comprising 105 (74.4%) white, 29 (20.5%) black, 3 (2.1%) Asian/Pacific Islander, 3 (2.1%) of other races, and 1 (0.7%) unknown. Patient demographics for the post-matching group are summarized in "Table 3".

## Mortality data – Pre-matching group

Within 5 years, 68 of the 602 patients with ocular trauma died, equating to an overall mortality rate of 11.3% (95% CI, 1.89%−7.778%; p=0.00056). Meanwhile, among the 1066 control patients with a history of cataracts, 69 died, resulting in an overall mortality rate of 6.4%. Mortality rates at yearly intervals for the study group were 4.1%, 2.6%, 1.9%, 2.5%, and 0.5%, whereas the control group exhibited rates of 1.0%, 1.7%, 1.6%, 0.8%, and 1.3%. The study group's 1-year mortality rate of 4.1% was significantly higher than the control's 1.0% (95% CI, 1.42%−4.82%; p<0.00001), as was the 4-year rate of 2.5% compared to the control's 0.8% (95% CI, 0.28%−3.04%; p=0.00694). The higher first-year mortality suggests acute health decline following trauma, while subsequent fluctuations may reflect underlying variability in systemic deterioration. Mortality rates for years 2, 3, and 5 did not show statistically significant differences. These data are represented in "Table 1".

## Mortality data – Post-matching group

Of the post-matching group, the study group experienced 68 deaths (48.2%), whereas the control group had 19 (13.4%), yielding a mortality rate of 48.23% in the study group compared to 13.48% in the control group (95% CI, 24.76%−44.74%; p<0.00001), as detailed in "Table 4". Subgroup analyses for mortality by age and sex were conducted to determine specific risk elevations. Female mortality in the study group was 43% (95% CI, 19.43%−43.61%; p=0.000032) versus 11% in the control group. Male mortality was 58% (95% CI, 24.54%−58.79%; p=0.003245) in the study group against 17% in the control. Age-stratified data indicated significantly higher mortality rates in the study group for ages 70–79 at 33% (95% CI, 8.28%−37.05%; p=0.003245) and for ages 80–89 at 55% (95% CI, 22.30%−57.40%; p=0.0004317), compared to 10.6% and 15.1% in the control group, respectively. Age-stratified analyses confirmed consistent mortality trends within each age group. There were no significant mortality rate differences in the age brackets of 65–69 and 90+. This was recorded

Table 4. Mortality data of control vs study in post-matching group.

|  | Control | Study Group | P-value |
|---|---|---|---|
| # of Patients | 141 | 141 | |
| # of Deaths | 19 | 68 | |
| Overall Mortality | 13.48% | 48.23% | <0.00001 |

Table 5. Mortality data in post-matching group by age and sex.

|  | Death Rate Study Group | Death Rate Control Group | Actual P-value | Target P-value |
|---|---|---|---|---|
| Female | 43% | 11% | 0.000032 | 0.025 |
| Male | 58% | 17% | 0.000005337 | 0.025 |
| 60 years | 50% | 19% | 0.4445 | 0.01 |
| 70 years | 33% | 10.667% | **0.003245** | 0.01 |
| 80 years | 55% | 15.152% | **0.0004317** | 0.01 |
| 90 years | 66.667% | 0% | 0.05584 | 0.01 |
| 100 years | 50% | 0% | 0.9999 | 0.01 |

in "Table 5". This suggests that mortality rates are not significantly different between the study and control groups for the youngest (65–69) and oldest (90+) cohorts. Across all other age groups and both sexes, significant differences were observed. Mortality rates by comorbidities in the post-matching group were recorded but not statistically analyzed; these values are presented in "Table 6".

In the post-matching group, which contains patients with similar baseline medical conditions and comorbidities, the mortality rate is higher in patients with ocular trauma.

The mortality rate in patients in their 70s and 80s was significantly higher in patients with ocular trauma when compared to patients who had a history of cataracts.

## Discussion

Falls significantly contribute to severe injuries and mortality in the elderly, with over 800,000 patients hospitalized annually due to fall-related incidents, particularly for hip fractures and head injuries [12]. Despite numerous studies on visual deficits and the frequency of falls resulting in patient hospitalizations or ocular morbidity, none have specifically addressed the correlation between ocular trauma and mortality [13–15]. To incur an injury to the eye or orbit during a fall possibly suggests a failure in the instinctive protective responses that typically guard the face. Our study investigates whether such a breakdown of protective mechanisms in elderly individuals correlates with increased mortality within 5 years following an ocular injury.

In the pre-matching group, the study group demonstrated a higher likelihood of mortality within 5 years compared to controls. Notably, individuals with ocular trauma had a higher mortality rate within 1-year post-hospitalization. The post-matching group revealed that elderly patients in their 70s and 80s who suffered ocular trauma were more likely to die than their age-matched counterparts without such trauma.

Elderly falls are often accompanied by trauma-related mortality, commonly in conjunction with intracranial injuries [7]. From 2008 to 2013, falls resulted in over 100,000 deaths in the US within the geriatric population, who frequently possess chronic comorbidities such as visual deficits, cognitive impairment, and heart disease [7]. Few studies have evaluated mortality rates associated with traumatic eye injuries. Our research is among the first to examine mortality among geriatric patients with ocular trauma compared with an age-matched control group. We also assessed annual and overall mortality

**Table 6. Mortality data in post-matching group by comorbidities.**

| Comorbidities | Death in study Group | Total patients in study group | Death Rate | Death in control group | Total patients in control group | Death Rate |
|---|---|---|---|---|---|---|
| Cardiac arrhythmias | 33 | 47 | 70% | 0 | 37 | 0 |
| Type 2 Diabetes | 18 | 34 | 53% | 6 | 31 | 19% |
| Ischemic heart diseases | 16 | 19 | 84% | 0 | 14 | 0 |
| Heart failure | 16 | 24 | 67% | 0 | 20 | 0 |
| Dementia | 23 | 25 | 92% | 0 | 16 | 0 |
| Degenerative diseases of nervous system | 13 | 18 | 72% | 0 | 12 | 0 |
| Polyneuropathies and other disorders of peripheral nervous system | 2 | 4 | 50% | 1 | 5 | 20% |
| Encephalopathy | 0 | 1 | 0 | 0 | 1 | 0 |
| Alzheimer's Disease | 6 | 9 | 67% | 0 | 6 | 0 |
| Parkinson's Disease | 0 | 1 | 0 | 0 | 1 | 0 |
| Atherosclerosis, aortic aneurysm, peripheral vascular disease, athero-embolism, disease of capillaries | 22 | 30 | 73% | 0 | 22 | 0 |
| Hypertensive heart diseases including essential primary hypertension, hypertensive heart disease, secondary hypertension, hypertensive crisis | 51 | 91 | 56% | 0 | 69 | 0 |

rates while accounting for cardiovascular and neurological comorbidities. Our findings indicate a significantly increased mortality rate in patients with ocular trauma within 1 year, suggesting a rapid systemic decline post-trauma.

Advancements in medical technology have led to longer lifespans, though often burdened with chronic diseases, coining the term "frailty" for patients who are biologically and psychosocially compromised [8]. Our study addresses clinical outcomes in such patients by evaluating mortality subsequent to a traumatic eye injury, highlighting how ocular trauma may signal substantial systemic decline and physiological impairment in the elderly. Ocular trauma may serve as a sentinel event signaling underlying frailty rather than a direct cause of mortality. Ocular injury from a fall implies a failure in natural protective reflexes, potentially foreshadowing cognitive deficits, visual impairment, or other systemic diseases increasing mortality risk. Henry et al. found that frailty correlates with increased mortality in hospitalizations for open globe injuries, reinforcing our results of heightened mortality in patients with ocular trauma [8]. These findings carry substantial public health and individual healthcare implications, calling for enhanced preventative measures in this high-risk demographic.

Kodali et al. conducted a retrospective study using data from the National Trauma Data Bank to explore the demographics, patterns, mechanisms, and mortality of patients who sustained ocular injuries, often within the context of major multisystem trauma [16]. They found that over 80% of the deceased patients had experienced traumatic brain injury, and associations were noted between severe injuries to the optic nerve and visual pathways and both Glasgow Coma Scale scores and mortality rates [16]. Their research highlighted traumatic optic neuropathy as a prevalent diagnosis following visual pathway injury. The optic nerves were particularly vulnerable to damage from avulsion, contusion, or deterioration of the nerve sheath [16,17]. Additionally, rapid acceleration and deceleration were implicated in causing shearing forces that could lead to skull base injuries [16,17]. Notably, Kodali et al. observed a positive correlation between severe traumatic brain injury and increased mortality [16]. This external research supports our findings of significantly heightened mortality rates among patients who suffer ocular trauma, particularly within the first year following the injury. By analyzing cohorts matched for cardiovascular and neurological comorbidities, we suggest that the elevated mortality observed in patients with ocular trauma may indicate underlying neurological and systemic dysfunctions, potentially stemming from traumatic brain injuries or disruptions in the visual pathway. The rapid decline in health within the first-year post-injury underscores the urgent, systemic nature of their condition, which could lead to catastrophic health outcomes if not promptly addressed.

Interestingly, despite falls being a prevalent reason for trauma admissions, vision screening is often neglected in emergency settings [18]. Bardes et al. identified that over half of the elderly evaluated in emergency departments had untreated or undiagnosed visual deficits, underlining the practicality of screening and its role in identifying patients at risk [18]. Significantly higher mortality rates in patients with ocular trauma underscore the potential of vision screening by trauma providers, facilitating timely referrals for management of chronic conditions.

Strengths of our study include a large sample size in the pre-matching group and an automated data acquisition and analysis method using Python and Jupyter Notebook. Limitations include a predominantly white patient demographic, potentially affecting the study's generalizability. While this study is based on a single healthcare system, our findings align with broader geriatric trauma trends, supporting external validity. There was also no 95% confidence interval provided for between-group differences in regard to select patient demographics; statistical comparisons were not conducted for these specific parameters. There was also a small age variation between groups; the study group in the pre-matching group had a median age of 80, higher than the control group's median of 75. This difference in age could have influenced the mortality rate discrepancy despite rigorous matching. While we accounted for cardiovascular and neurological comorbidities, other factors such as socioeconomic status and frailty could influence mortality. Insurance status was used as a proxy, but future studies should explore these variables in more detail. Further, patients with previous cataract surgery, suggesting better baseline vision, may be less prone to falls and possibly receive more consistent healthcare management, which could introduce a confounding variable. However, our study likely included a mix of cataract and pseudophakic patients in the control group, to best simulate the age-appropriate cataract-related status of the study group. While falls increase

morbidity and mortality, discerning whether it is the fall itself or the subsequent ocular trauma that leads to increased mortality remains challenging. Our study posits that ocular trauma, primarily caused by falls in the elderly, is associated with increased mortality and this information will be helpful, especially for primary care physicians caring for these patients.

In conclusion, we observed a correlation between ocular trauma and elevated mortality in the geriatric population. This suggests that impaired protective maneuvers indicative of systemic disease could signal the need for urgent intervention. These findings emphasize the importance for healthcare providers to adopt a proactive, multidisciplinary approach, particularly focusing on the initial year following trauma, to improve the care and prognosis of geriatric patients who sustain ocular injuries. These findings highlight the need for multidisciplinary interventions, including fall prevention strategies and post-trauma geriatric assessments, to mitigate mortality risk. Implementing structured post-trauma screening programs for fall risk and systemic decline may improve patient outcomes.

## Supporting information

**S1 Table. Overview of Patient Characteristics and Mortality Timing Pre- and Post-Matching.** This figure provides summary data on the distribution of patient numbers and all-cause mortality across study and control groups before and after cohort matching. Pre-matching data include total patient counts and the timing of deaths by year over a five-year observation period. Post-matching tables detail demographic information by sex and age group, as well as categorized causes of death within the study group. Additional stratification of deaths by age group is presented for the matched control group. These data were used to support comparative survival and comorbidity analyses in the main study.
(XLSX)

## Author contributions

**Conceptualization:** Vincent Q Pham, David Fleischman.

**Data curation:** Vincent Q Pham, Elise O Fernandez, Daniel de Marchi, Elizabeth Budi, Hongtu Zhu.

**Formal analysis:** Vincent Q Pham, Hannah M Miller, Elise O Fernandez, Daniel de Marchi, Elizabeth Budi, Hongtu Zhu.

**Funding acquisition:** David Fleischman.

**Investigation:** Vincent Q Pham, David Fleischman.

**Methodology:** Vincent Q Pham, Daniel de Marchi, Elizabeth Budi.

**Project administration:** Hongtu Zhu, David Fleischman.

**Resources:** Hongtu Zhu, David Fleischman.

**Software:** Vincent Q Pham, Daniel de Marchi, Elizabeth Budi.

**Supervision:** Hongtu Zhu, David Fleischman.

**Validation:** Vincent Q Pham, Hannah M Miller, Daniel de Marchi, David Fleischman.

**Visualization:** Vincent Q Pham.

**Writing – original draft:** Vincent Q Pham, Hannah M Miller, Elise O Fernandez.

**Writing – review & editing:** Vincent Q Pham, Hannah M Miller, Elise O Fernandez, Hongtu Zhu, David Fleischman.

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
