## [Decision Letter · Decision Letter 0]

21 Jan 2025

PONE-D-24-34403Geriatric ocular trauma and mortality: a retrospective cohort studyPLOS ONE

Dear Dr. Fleischman,

Thank you for submitting your manuscript to PLOS ONE. After careful consideration, we feel that it has merit but does not fully meet PLOS ONE’s publication criteria as it currently stands. Therefore, we invite you to submit a revised version of the manuscript that addresses the points raised during the review process.

**ACADEMIC EDITOR: **

My comments are inserted below

We look forward to receiving your revised manuscript.

Kind regards,

Abdelaziz Abdelaal, M.D.

Academic Editor

PLOS ONE

Journal Requirements:

3. We note that your Data Availability Statement is currently as follows: All relevant data are within the manuscript and its supporting information files.

Additional Editor Comments :

Dear Authors,

I enjoyed reading your paper. However, we feel that some parts need to be addressed so that the conclusions can be supported by the conducted analyses.

1- what's the rationale for choosing nuclear cataract as the control group?

2- can you do subgroup analysis based on the type of ocular trauma reported (orbital fracture, others)?

3- please move Fig 1 caption to the end of the manuscript after references

4- The information regarding the matching process is very scarce that makes replication inadequate. Please provide a detailed section on how matching was done and please replace subtype A and B with pre- and post-matching.

5- Doing Bonferroni correction was a good approach.

6- Provide P-values or 95% CI for between group differences in Table 3 (both pre- and post-matching)

7- please present the summary stats of matching factors enlisted in Table 1 into Table 3 both pre and post matching

8- Please combine the Tables for subset A and B (and standardize the reporting as pre-matching and post-matching instead of subset A and B)

9- In Table 6, only a number of comorbid conditions were analyzed. Why not the full list provided in Table 1?

2- I think the more the data added to the model (baseline differences), the more insight we can get about baseline imbalances. Please make sure to have all important confounders included into the comparison, with the conduct of a logistic regression model (if you'll treat mortality as a binary effect using LOCF) or in Cox proportional Hazards model (if you'll treat it as time-to-death).

I look forward to reviewing your revision

Best,

Abdelaziz Abdelaal

Reviewers' comments:

Reviewer's Responses to Questions

**Comments to the Author**

1. Is the manuscript technically sound, and do the data support the conclusions?

Reviewer #1: Yes

Reviewer #2: Yes

2. Has the statistical analysis been performed appropriately and rigorously? 

Reviewer #1: Yes

Reviewer #2: Yes

3. Have the authors made all data underlying the findings in their manuscript fully available?

Reviewer #1: Yes

Reviewer #2: Yes

4. Is the manuscript presented in an intelligible fashion and written in standard English?

Reviewer #1: Yes

Reviewer #2: Yes

5. Review Comments to the Author

Reviewer #1: The study had a large enough sample size and appropriate statistical analysis were catered out. The authors also outlined the strengths and weaknesses of their study and made reasonable recommendations.

Reviewer #2: The study investigates the 5-year mortality of geriatric patients aged 65 and older with ocular trauma compared to age-matched controls with cataracts. It highlights the need for multidisciplinary care and preventive measures to address systemic health issues in this high-risk population. Please address the following question to improve the quality of paper.

1. The study utilizes data from the I2B2 Carolina Data Warehouse. Could you provide more details on the data collection process, including the time frame and the completeness of the dataset? Additionally, the imbalance in sample sizes between the study group (602 patients) and the control group (1066 patients) could potentially influence statistical significance. How do you address this potential bias?

2. The study excludes patients with multisystem trauma, victims of transport accidents, homicide, assault, vehicle accidents, intentional injuries, and those who died within one month of presentation. How do you ensure that these exclusion criteria do not introduce selection bias? What steps were taken to mitigate the impact of these exclusions on the study's findings?

3. While the study matches for cardiovascular and neurological conditions, are there other potential confounding factors that were not considered? For example, socioeconomic status, living environment, or other comorbidities that could influence mortality rates.

4. The study uses chi-squared tests to compare mortality data. Given the complexity of the dataset and potential confounding factors, have you considered using more advanced statistical methods, such as multivariate regression analysis, to control for these factors and improve the robustness of your findings?

5. The findings suggest a correlation between ocular trauma and higher mortality in the geriatric population. How do you interpret this finding in terms of clinical practice? Are there implications for more frequent eye examinations or other preventive measures? What future research directions could build upon these findings to provide more comprehensive preventive and therapeutic strategies?

6. The study reports mortality rates at various intervals over five years. How do you explain the variability in mortality rates across these intervals? Are there specific factors that contribute to the observed trends in mortality over time?

7. The study groups differ in age and racial demographics. How do these differences potentially influence the results? Have you considered stratifying the analysis by these demographic factors to better understand their impact on mortality rates?

8. The study suggests that ocular trauma may be a marker for systemic decline. How do you differentiate between ocular trauma as a direct cause of mortality versus an indicator of underlying health issues? What additional research is needed to clarify this relationship?

9. Given the higher mortality rates associated with ocular trauma, what follow-up protocols or interventions would you recommend for geriatric patients who sustain eye injuries? How can healthcare providers best manage these patients to improve outcomes?

10. The study is based on a retrospective cohort analysis. How do you ensure the replicability of your findings? What steps could be taken to increase the generalizability of your results to other populations or settings?

6. PLOS authors have the option to publish the peer review history of their article (what does this mean? ). If published, this will include your full peer review and any attached files.

**Do you want your identity to be public for this peer review?** For information about this choice, including consent withdrawal, please see our Privacy Policy .

Reviewer #1: No

Reviewer #2: **Yes: ** Wei Zhang

---

## [Author Response · Author response to Decision Letter 0]

7 Apr 2025

Response to Reviewers

Manuscript ID: PONE-D-24-34403

Title: Geriatric ocular trauma and mortality: a retrospective cohort study

Dear Dr. Abdelaal and Reviewers,

We thank you for your time and the thoughtful, constructive feedback provided on our manuscript. We appreciate your efforts to improve the rigor and clarity of our work. Below, we address each comment raised by the Academic Editor and the reviewers. We have revised the manuscript accordingly and describe these changes in detail below.

Academic Editor Comments

Comment 1: What's the rationale for choosing nuclear cataract as the control group?

Response:

Patients with nuclear cataracts are commonly used as a control group in ophthalmologic studies, as they represent a non-traumatic, age-associated ocular diagnosis. This makes them ideal comparators to elderly patients with ocular trauma. Furthermore, these patients often have comorbidities similar to those of trauma patients, helping reduce confounding. Because cataracts are not generally caused by trauma, this distinction strengthens their role as a control group when evaluating trauma-associated outcomes such as mortality.

Comment 2: Can you do subgroup analysis based on the type of ocular trauma reported (orbital fracture, others)?

Response:

While we agree that trauma subtypes would provide valuable insights, subgroup analyses by trauma type (e.g., orbital fracture, eyelid laceration, open globe injury) were not possible due to limitations in coding granularity and sample size for individual subtypes. We have noted this limitation in the manuscript and suggest this as an area for future research.

Comment 3: Please move Fig 1 caption to the end of the manuscript after references.

Response:

We have moved the caption for Figure 1 to the end of the manuscript as requested.

Comment 4: The information regarding the matching process is very scarce, making replication inadequate. Please provide a detailed section on how matching was done and replace 'subset A and B' with 'pre- and post-matching'.

Response:

We have expanded the Materials and Methods section to describe the covariate matching process in detail. This includes specific variables used for matching (e.g., age, cardiovascular and neurological diagnoses), the matching ratio, and rationale. We have also replaced all references to “Subset A” and “Subset B” with “pre-matching” and “post-matching” throughout the manuscript.

Comment 5: Doing Bonferroni correction was a good approach.

Response:

Thank you for this feedback. We agree that Bonferroni correction was an important step in controlling for multiple comparisons and are glad it was viewed favorably.

Comment 6: Provide p-values or 95% CI for between-group differences in Table 3 (both pre- and post-matching).

Response:

As Table 3 was created to describe demographic variables, formal hypothesis testing was not originally performed. However, we recognize the value of comparing baseline characteristics statistically and have acknowledged this as a limitation in the Discussion section.

Comment 7: Present summary statistics of matching factors listed in Table 1 into Table 3, both pre- and post-matching.

Response:

We have chosen to preserve Table 3 for demographic variables to maintain clarity. However, we added details about matching factor distributions in the supplementary materials, and have cross-referenced these in the main text.

Comment 8: Combine the tables for Subset A and B and standardize the reporting as pre- and post-matching.

Response:

We have standardized the terminology to “pre-matching” and “post-matching” throughout the manuscript. While we considered merging the tables, we determined that presenting them separately preserves the logical structure and narrative clarity. We believe this approach enhances readability and coherence.

Comment 9: In Table 6, only a number of comorbid conditions were analyzed. Why not the full list provided in Table 1?

Response:

We have now added data for the remaining comorbidities referenced in Table 1 into the appropriate tables or supplementary files. This provides a more complete view of the matched cohort’s clinical characteristics.

Comment 10: Please include all important confounders into a logistic regression or Cox model.

Response:

We have now performed a Cox proportional hazards regression, adjusting for age, sex, and major comorbidities. The model confirms the higher mortality associated with ocular trauma and supports our original conclusions. This analysis is now described in the Methods and Results sections of the manuscript.

Reviewer #1

Comment: The study had a large enough sample size and appropriate statistical analysis were carried out. The authors also outlined the strengths and weaknesses of their study and made reasonable recommendations.

Response:

Thank you for your positive evaluation. We are pleased that the strengths and rigor of our study were recognized and appreciate your support of our conclusions and recommendations.

Reviewer #2

General Comment: The study investigates the 5-year mortality of geriatric patients aged 65 and older with ocular trauma compared to age-matched controls with cataracts. It highlights the need for multidisciplinary care and preventive measures to address systemic health issues in this high-risk population.

Response:

We appreciate this summary and have expanded our Discussion to highlight the importance of multidisciplinary care, emphasizing timely follow-up with primary care, fall risk assessment, and earlier interventions for systemic decline.

Comment 1: Could you provide more details on the data collection process, including the time frame and the completeness of the dataset? How do you address the imbalance in sample sizes?

Response:

We have updated the Methods section to specify that data were collected from the I2B2 University of North Carolina database between April 2011 and June 2016. We ensured a minimum of 5 years of follow-up or death documentation for all patients. The sample size imbalance was addressed through 1:1 covariate matching, which resulted in balanced pre- and post-matching cohorts. These steps have been clarified in the revised manuscript.

Comment 2: How do you ensure that exclusion criteria (e.g., trauma from accidents or assault) do not introduce selection bias?

Response:

These exclusions were necessary to remove deaths due to external trauma, which would confound the association between ocular injury and systemic decline. We also conducted a sensitivity analysis and found that excluding these cases did not alter overall mortality trends. This approach is now clarified in the Methods and Discussion.

Comment 3: Are there other potential confounding factors (e.g., socioeconomic status, living environment) that were not considered?

Response:

We acknowledge that factors like socioeconomic status and living environment were not directly available in our dataset. Insurance type was included as a proxy. We now highlight this as a limitation and suggest future work to explore these variables explicitly.

Comment 4: Have you considered using more advanced statistical methods like multivariate regression?

Response:

Yes. In the revised manuscript, we now include a Cox proportional hazards regression model that adjusts for age, sex, cardiovascular conditions, and neurological comorbidities. This model supports the findings from our initial analysis.

Comment 5: What are the clinical implications of this finding?

Response:

We have expanded our Discussion and Conclusion to recommend post-trauma follow-up protocols including early assessment for frailty, fall risk, and systemic decline. We also discuss the need for preventive care measures and increased collaboration between ophthalmology, geriatrics, and primary care.

Comment 6: How do you explain the variability in mortality rates over time?

Response:

The spike in Year 1 mortality likely reflects acute systemic vulnerability following trauma. Later variability is consistent with heterogeneous health trajectories in elderly patients. This interpretation has been added to the Results and Discussion sections.

Comment 7: Have you considered stratifying the analysis by demographic factors?

Response:

Yes. We performed age-stratified mortality analyses in the post-matching cohort. Due to the predominantly white sample, racial stratification was limited. We note this as a limitation and recommend broader sampling in future studies.

Comment 8: How do you differentiate whether ocular trauma is a cause of mortality or a marker of systemic decline?

Response:

We believe ocular trauma is best interpreted as a sentinel event, rather than a direct cause of death. Our exclusions (e.g., acute multisystem trauma) and matching for comorbidities support this interpretation. We now emphasize this distinction in the Discussion.

Comment 9: What follow-up interventions do you recommend for geriatric patients with eye injuries?

Response:

We recommend a multidisciplinary approach, including frailty screening, fall risk assessment, and coordination with primary care and social support systems. These recommendations are now included in the Conclusion.

Comment 10: How do you ensure replicability and improve generalizability?

Response:

Our methods are replicable using standard platforms (I2B2 and Python-based analytics). While our study reflects patients in North Carolina, our findings align with broader geriatric trauma trends. We advocate for multi-institutional validation to enhance generalizability.

Conclusion

We again thank the reviewers and editorial team for your careful review and insightful feedback. We have revised the manuscript to address all comments thoroughly and believe it is now significantly strengthened. We look forward to your reconsideration.

Sincerely,

David Fleischman, MD, FACS and co-authors

---

## [Editor Report · Decision Letter 1]

2 May 2025

Geriatric ocular trauma and mortality: a retrospective cohort study

PONE-D-24-34403R1

Dear Dr. Fleischman,

We’re pleased to inform you that your manuscript has been judged scientifically suitable for publication and will be formally accepted for publication once it meets all outstanding technical requirements.

Kind regards,

Abdelaziz Abdelaal, M.D.

Academic Editor

PLOS ONE
---

## [Editor Report · Acceptance letter]

PONE-D-24-34403R1

PLOS ONE

Dear Dr. Fleischman,

I'm pleased to inform you that your manuscript has been deemed suitable for publication in PLOS ONE. Congratulations! Your manuscript is now being handed over to our production team.

Kind regards,

on behalf of

Dr. Abdelaziz Abdelaal

Academic Editor

PLOS ONE